# *Hypericum* Essential Oils—Composition and Bioactivities: An Update (2012–2022)

**DOI:** 10.3390/molecules27165246

**Published:** 2022-08-17

**Authors:** Maria-Eleni Grafakou, Christina Barda, George Albert Karikas, Helen Skaltsa

**Affiliations:** 1Department of Pharmacognosy & Chemistry of Natural Products, Faculty of Pharmacy, School of Health Sciences, National & Kapodistrian University of Athens, 15771 Athens, Greece; 2Institute of Pharmaceutical Sciences, Department of Pharmacognosy, University of Graz, Beethovenstraße 8, 8010 Graz, Austria; 3Department of Biomedical Sciences, University of West Attica, 12243 Athens, Greece

**Keywords:** *Hypericum*, essential oil, chemical composition, in vitro, in vivo, biological activity

## Abstract

*Hypericum* genus, considered to comprise over 500 species that exhibit cosmopolitan distribution, has attracted human interest since ancient times. The present review aims to provide and summarize the recent literature (2012–2022) on the essential oils of the title genus. Research articles were collected from various scientific databases such as PubMed, ScienceDirect, Reaxys, and Google Scholar. Scientific reports related to the chemical composition, as well as the in vitro and in vivo pharmacological activities, are presented, also including a brief outlook of the potential relationship between traditional uses and *Hypericum* essential oils bioactivity.

## 1. Introduction

The genus *Hypericum* L. (Hypericaceae) includes more than 500 taxa with a worldwide distribution classified into 36 taxonomic sections [1]. The botanical name derives from the Greek word hypericon (“υπέρ εικόνα” meaning above the icon), suggesting its use against the evil eye. The use of *Hypericum* has been reported even during classical antiquity by Hippocrates [2], Dioscorides [3], and later in the Medieval era by Nikolaos Myrepsos [4,5]. The World Health Organization (WHO) reported that 80% of the world’s population uses medicinal plants for primary health needs [6]. Furthermore, research in the field of natural products has gained great attention in the past few decades, a trend expected to continue in the coming years. Considering the current decrease in new drugs introduced to the market, as well as nature’s high potential for yielding therapeutically relevant bioactive compounds, plant metabolites are emerging as new lead structures for the development of novel drugs for treating various diseases [7,8,9].

In this context, we should mention that several *Hypericum* spp. are used throughout the world in folk medicine, as astringent, febrifuge, diuretic, antiphlogistic agent, analgesic, and antidepressant agents [10]. In the 18th and 19th centuries, European and American physicians used *Hypericum* in the treatment of various health problems such as headaches, bed wetting, burns, puncture wounds, vertigo, hyperhidrosis, melancholy, and paranoia. In addition, modern medical research has shown that *H. perforatum* (with the common name St. John’s wort) is an effective herbal medicine for the treatment of mild to moderate depression [ΕΜA, herbal medicine with well-established use], while a monograph also mentions wound-healing properties [ΕΜA, herbal medicine with traditional use]. The active constituents of the genus mainly belong to the groups of phloroglucinols, napthodianthrones, xanthones, and flavonoids [11]. These metabolites display a wide range of biological activities and attract the interest of the scientific community; apart from the well-established antidepressant activity [12], many studies describe cytotoxic, antimicrobial, and anti-inflammatory effects [13,14,15,16].

Essential oils (EOs) represent an interesting mixture of volatile compounds and are reported to possess strong antimicrobial, antioxidant, and antiangiogenetic activities, although *Hypericum* species are generally classified as EO-poor plants [17]. According to the European Pharmacopeia, EOs (Aetherolea) are odorous products, usually of complex composition, obtained mainly by steam distillation. These complex mixtures mostly consist of mono- and sesquiterpenes, in the form of hydrocarbons or oxygenated derivatives. Other substances that could be co-extracted with EOs from different plants are diterpenes, phenols, fats, coumarins, anthraquinones, certain alkaloids, and several compounds derived during the distillation process (artifacts). EOs from diverse plants have a plethora of uses in the cosmetic, pharmaceutical, and food industries [18].

Ten years have passed since the last reviews were published on *Hypericum* spp. EOs [17,19,20]. Since then, 71 papers have been published describing relevant research work [10,14,15,16,18,21,22,23,24,25,26,27,28,29,30,31,32,33,34,35,36,37,38,39,40,41,42,43,44,45,46,47,48,49,50,51,52,53,54,55,56,57,58,59,60,61,62,63,64,65,66,67,68,69,70,71,72,73,74,75,76,77,78,79,80,81,82,83,84,85,86], 34 of them referring to biological activities [20,22,24,32,34,38,39,40,41,42,46,49,52,54,57,58,60,62,63,64,65,66,67,68,69,70,71,72,74,77,79,82,83,86] of which antiviral, antimalarial, cytotoxic, neuroprotective, tyrosinase inhibition, immunomodulatory, anti-angiogenic, hepatoprotective, and wound-healing effects had not been tested earlier, thus later data are not described in previous reviews. As a result, we updated the latest information regarding *Hypericum* EOs’ chemical composition and biological activities and we attempted to bridge the potential relationship between traditional uses and *Hypericum* EOs.

## 2. Methodology

A selection of the relevant data was performed through a search using the keyword “Hypericum essential oil” in “PubMed”, “Scopus”, “Reaxys”, and “Google Scholar” databases. Approximately 100 articles were found. In the present review, the search terms “Hypericum perforatum essential oil”, “Hypericum biological activities”, “Hypericum pharmacological activities”, “Hypericum essential oil biological”, and “Hypericum essential oil pharmacological” were further used. In total, 73 publications describing the EOs and their biological activities were used, excluding articles solely on botany and agronomy. The major constituents of the EOs are categorized by species in Table 1, and similarly, the reported biological activities are categorized by species in Table 2 and further discussed by pathology in Section 3.2. Plant taxonomy was validated using the databases of the International Plant Names Index (IPNI). Information about folk uses and botany was obtained from published books and academic papers solely in the English language.

## 3. Results and Discussion

Aid products of Hypericum plants are being sold around the world and comprise an important portion of the market. In fact, such commodities are widely consumed as supplements to maintain and improve human health, thus the investigation of Hypericum plants (including different preparations and also EOs) in terms of chemical composition and biological effects is of high importance. Regarding Hypericum EOs, several major constituents have been reported from these plants with relatively limited distribution among other genera, with potential use in food and cosmetic industries [17].

Hypericum is generally considered an EO-poor genus, with very low yields < 1%, though the literature indicates in some cases higher EO yields, similar to other genera, up to 3% [21]. Moreover, it is reported that the EO content during the full-bloom stage vs. the pre-bloom or fruiting stage is higher [17]. It is also estimated that the careful selection of inflorescences and leaves instead of the total aerial part also leads to higher EO yields of up to 13 *v*/*w*% using hydro-distillation [22]. Hydro-distillation is by far the most common method used to obtain EOs, which is also proposed by EMA monographs and European Pharmacopoeia; however, to date, other techniques have been introduced in order to improve the extraction efficiency and control the chemical composition of the plant material, including liquid or supercritical extraction, solid-phase microextraction, and ultrasound-assisted headspace [17,23,24]. Moreover, other biotechnological tools have been applied, such as callus culture, which produces biomass on a large scale and could be used as a good experimental system for further research on essential oil production. Calli essential oil cultivation enabled the selection of a desired compound or group of compounds with specific aromas or activities as a response to chemical elicitors that stimulate biotic and abiotic stress in vitro [25,26]. Nevertheless, Gas Chromatography (GC) is, by all means, the ‘golden standard method’ in the chemical analysis of EOs, especially enforced with the aid of GC-MS (Mass Spectrometry) and GC-FID (Flame Ionization Detector) for both the identification and quantification of the content as well as the composition variations, regardless of the extraction protocol [17,22,23,24].

Most of the studies on *Hypericum* EOs have been conducted with single plant materials from wild populations and without repetition, though there are several taxa that have been thoroughly investigated, such as *H. perforatum*, *H. scabrum*, *H. perfoliatum*, and *H. triquertifolium*. During the last decade, several taxa were investigated for the first time, specifically *H. kotschyanum*., *H. salsugineum*, *H. uniglandulosum* [27], *H. silenoides,* and *H. philonotis* [28], *H. aviculariifoliu* ssp. *depilatum* var. *depilatum* [29,30], *H. empetrifolium* ssp. *empetrifolium* [31], *H. reflexum*, *H. canariense* and *H. grandifolium* [32], *H. asperulum* [33], *H. gaiti* [34], *H. pruinatum* [30], *H. origanifolium* [35], *H. laricifolium* [36], *H. japonicum* [37], *H aegypticum* subdp. *webii* [38], *H. amblyocalyx* and *H. jovis* [39], *H. hemsleyanum* [40], *H. hookerianum* and *H. bellum* [24], *H. rochelii* and *H. umbellatum* [41], and *H. hyssopifolium* ssp. *elongatum* var. *microcalycinum* [42]. In addition, the aroma of the berry-like fruits of *H. androsamemum* was chemically investigated [43].

### 3.1. Chemical Constituents of Hypericum spp. EOs

*Hypericum* EOs include the following main constituents: The monoterpene hydrocarbons α- and β-pinenes, the sesquiterpene hydrocarbons E-caryophyllene and germacrene D, and the oxygenated sesquiterpenes spathulenol and caryophyllene oxide, while in some cases, the major compounds are n-alkanes, such as undecane and n-nonane (Table 1).

**Table 1 molecules-27-05246-t001:** Literature survey (2012–2022) on essential oils from *Hypericum* spp.

*Hypericum* spp.	Plant Origin	Main Ingredients of EOs	Reference
*H. aegypticum* ssp. *webbii* (Spach) N. Robson	Greece	α-pinene (63.4–68.5%), β-pinene (16.9–17.0%) (two collection points)	[38]
*H. amblyocalyx* Coustur. & Gand	Greece	β-elemene (17.4%), β-selinene (10.5%), α-pinene (10.2%), E-caryophyllene (8.8%), α-selinene (8.7%)	[39]
*H. ascyron* L. (syn. *H*. *hemsleyanum* H.Lév. & Vaniot)	China	osthole (35.6%)	[40]
*H. asperulum* Jaub. & Spach	Iran	γ-muurolene (13.1%), α-pinene (12.2%), germacrene D (11.3%), β-caryophyllene (9.8%), spathulenol (7.2%)	[33]
*H. bellum* H.L.Li	China	curdione (30.9%), eicosyl nonyl ether (15.5%), but-3-yn-2-yl ester of undec-10- ynoic acid (9.4%), palmityl palmitoleate (9.3%)	[24]
*H. canariense* L.	Canary Islands	n-nonane (44.3%), (E)-caryophyllene (7.9%), β-pinene (7.7%)	[32]
*H. capitatum* Choisy	Turkey	spathulenol (12.9%), iso-longifolene (11.2%)	[56]
*H. saturejifolium* Jaub. & Spach (syn. *H. confertum* Choisy)	Turkey	germacrene D (30.2%)	[42]
	Turkey	α-pinene (7.8%), γ-muurolene (7.2%), δ-cadinene (6.5%)	[53]
*H. empetrifolium* Willd.	Greece	α-pinene (37.5%), iswarane (30.5%)	[21]
	Greece	α-pinene (19.0%), germacrene D (12.5%), β-pinene (8.7%), E-caryophyllene (5.3%)	[39]
	Turkey	allo-aromodendrene (24.7%), α-pinene (14.7%), β-pinene (10.7%), α-terpineol (7.7%)	[55]
*H. empetrifolium* Willd. ssp. *empetrifolium*	Greece	(E)-β-farnesene (29.5%), α-pinene (18.7%), (E)-β-caryophyllene (10.1%)	[31]
*H. gaitii* Haines	India	α-pinene (69.5%), β-caryophyllene (10.5%), sabinene (5.6%), myrcene (3.0%), geranyl acetate (2.0%)	[34]
*H. grandifolium* Choisy	Canary Islands	n-nonane (42.3%), (E)-caryophyllene (24.2%)	[32]
*H. helianthemoides* (Spach) Boiss.	Iran	α-pinene (31.9%), (E*)-β-ocimene* (12.5%), β-phellandrene (8.4%), β-pinene (6.3%), β-caryophyllene (5.7%), germacrene-D (4.3%)	[57]
*H. hircinum* L.	Turkey	α-pinene (88.3%)	[42]
	Greece	(E)-caryophyllene (65.87%)	[21]
*H. hircinum* L. ssp. *majus* (Aiton) N. Robson	Italy	cis-β-guaiene (29.3%), δ-selinene (11.3%), isolongifolan-7-α-ol (9.8%), (E)-caryophyllene (7.2%)	[54]
*H. hookerianum* Wight & Arn.	China	triacontane (26.4%), 1-iodotetracosane (20.6%), 2-methyl-2-decanol (14.8%), 2-(5-ethenyl-5-methyloxolan-2-yl) propan-2-yl ethyl carbonate (3.9%), aromadendrane (1.3%)	[24]
*H. humifusum* L.	Tunisia	α-pinene (27.8%), caryophyllene oxide (12.5%), β-pinene (11.5%), n-undecane (5.0%)	[60]
*H. japonicum* Thunb. ex Murray	India	2-methyl octane (24.9%), n-nonane (21.4%), (2Z)-nonenol (16.5%), n-decanal (8.2%), allo-aromadendrene epoxide (3.3%)	[37]
*H. jovis* Greuter	Greece	trans-calamenene (13.5%), α-selinene (8.3%), β-elemene (7.6%)	[39]
*H. kotschyanum* Boiss.	Turkey	α-pinene (14.4%), nonacosane (11.1%), hexadecanoic acid (9.2%), β-pinene (8.7%), spathulenol (6.3%), limonene (5.1%)	[27]
*Η. laricifolium* Juss.	Mérida-Venezuela	α-pinene (20.2%), verticiol (13.4%), 3-methyl-nonane (12.3%), 2-methyl-octane (9.6%), nonane (7.6%)	[36]
*H. lydium* Boiss.	Turkey	verbenone (22.2%), caryophyllene oxide (18.3%), α-eudesmol (11.3%), cis-linolool oxide (6.8%), β-selinene (6.3%)	[53]
	Turkey	α-pinene (58%), β-pinene (5.10%), β -myrcene (3.1%)	[51]
	Turkey	α-pinene (71.2%)	[52]
*H. maculatum* Crantz	Serbia	germacrene D (21.5%), nonane (6.5%), (E)-β-farnesene (5.3%), δ-cadinene (4.5%), ledol (4.4%)	[46]
*H. microcalycinum* Boiss. & Heldr. (syn. *H. hyssopifolium* Chaix ssp. *elongatum* (Ledeb.) Woron var. *microcalycinum*. (Boiss. & Heldr.) Boiss.)	Turkey	α-pinene (57.8%)	[42]
*H. orientale* L.	Turkey	β-selinene (37.1%), β-caryophyllene (9.7%), γ-muurolene (4.4%), cadinene (6.1%)	[53]
*H. origanifolium* Willd.	Turkey	α-selinene (19.6 or 18.7%), β-selinene (16.1 or 15.3%), γ-muurolene (4.6 or 4.7%), δ-cadinene (8.2 or 7.7%), spathulenol, 4.2 or 5.1%) (from leaves and flowers)	[35]
*H. origanifolium* var. *depilatum* (Freyn & Bornm.) N.Robson (syn. *H. aviculariifolium* ssp. *depilatum* (Freyn & Bornm.) N.Robson)	Turkey	α-pinene (52.1%), germacrene D (8.5%), β-pinene (3.6%)	[29]
*H. patulum* Thunb.	China	nonane 17.1–32.6% (undried and dried sample)	[58]
	Iran	β-pinene (30.2%), α-pinene (18.3%), limonene (8.4%), α-humulene (2.3%)	[59]
*H. perfoliatum* L.	Greece	γ-muurolene (8.5%), δ-cadinene (7.8%), γ-cadinene (5.3%), (E)-β-caryophyllene (6.6%), germacrene D (5.9%), n-undecane (4.2%)	[31]
	Kosovo	2-methyl-octane (1.1–15.5%), α-pinene (3.7–36.5%), β-caryophyllene (1.2–12.4%), caryophyllene oxide (3.3–17.7%), n-tetradecanol (3.6–10.4%) (different populations)	[61]
	Romania	α-pinene (30.9%), β-pinene (18.3%), caryophyllene (15.3%)	[62]
	Iran	2,6-dimethyl-heptane (6.3–36.1%), α-pinene (5.5–26.0%), γ-cadinene (0.0–22.6%), δ-cadinene (0.0–16.9%) (different populations)	[48]
	Iran	decane (59.6%), dodecane (12.9%), ethylcyclohexane (6.8%), 5-methylnonane (4.7%), 3-methylnonane (4.3%), tetradecane (3.8%)	[63]
	Iran	α-pinene (25.4%), α-amorphene (12.1%)	[64]
	Albania	caryophyllene oxide (31.0%), δ-selinene (10.5%), carvacrol (10.4%)	[65]
		β-pinene (24.9%), α-pinene (31.8%), caryophyllene (9.1%)	[66]
	Turkey	α-pinene (33.3%)	[42]
	Iran	α-pinene (12.5%), β-pinene (8.3%), undecane (7.0%), germacrene-D (6.9%)	[57]
	Tunisia	α-pinene (5.4%), β-selinene (8.9%), α-selinene (5.0%), 1-tetradecanol (10.2%)	[60]
	Albania	caryophyllene oxide (31.0%), δ-selinene (10.5%), carvacrol (10.4%)	[65]
	Iran	germacrene-D (15.2%), limonene (11.0%), β-caryophyllene (10.9%), α-pinene (10.7%), β-pinene (9.7%), germacrene-B (6.9%), α-guaiene (4.6%), β-farnesene (4.3%), spathulenol (2.5%), caryophyllene oxide (2.3%), δ-cadinene (2.1%), trans-ocimene (1.9%)	[23]
	USA Greece	(flowers) cis-p-menth-3-en-1,2-diol (9.1%), α-terpineol (6.1%), terpinen-4-ol (7.4%), limonen-4-ol (3.2%); (leaves) germacrene D (25.7%), β-caryophyllene (9.5%), terpinen-4-ol (2.6%)	[67]
		ishwarane (22.0%), α-himachalene (6.9%), α-pinene (6.4%), β-pinene (6.1%)	[22]
	China	(from Wuxic) docosyl heptyl ether (31.3%), pentyl tetracosyl ether (5.4%), 2-methyl-2-decanol (3.4%), heptacosane (1.3%) pentyl linoleate (1.0%); (from Wushan) docosyl heptyl ether (28.1%), 2-nonanone (6.8%), 2-methyl-2-decanol (5.0%), undecane (2.9%), linalyl oxide (2.3%), pentyl linoleate (2.2%)	[24]
*H. perforatum* L. ssp. *veronense* (Schrank) H. Lindb.	Croatia	α-pinene (16.6%), n-nonane (13.6%)	[49]
	Greece	α-selinene (14.6%), β-selinene (14.7%), (E)–β–caryophyllene (10.3%), α-pinene (7.5%), germacrene-D (5.52%)	[31]
*H. philonotis* Schltdl. & Cham.	Mexico	2-methyloctane (52.7%), n-nonane (35.9%), β-pinene (3.5%), 3-methyl-nonane (2.3%)	[28]
*H. pruinatum* Boiss. & Balansa	Turkey	β-selinene (15%), β-caryophyllene (8%), γ-muurolene (7%), α-selinene (6%), E-β-farnesene (4%), caryophyllene oxide (9%)	[30]
*H. pseudohenryi* N.Robson	China	heptacosane (2.7%), geranylgeraniol (1.9%), palmitic acid (1.8%)	[24]
*H. reflexum* L.	Canary Islands	α-pinene (3.3–16.7%), β-pinene (4.6–7.6%), n-undecane (9.7–17.6%), (E)-caryophyllene (4.9–8.2%), δ-cadinene (6.1–7.0%), α-cadinol (1.1–2.8%), caryophyllene oxide (1.4–1.6%) (from 2 collection sites)	[32]
*H. rochelii* Griseb. & Schenk	Serbia	n-nonane (24.7%), β-pinene (22.4%), germacrene D (7.5%), n-undecane (6.8%), α-pinene (5.8%)	[41]
*Η. rumeliacum* Boiss.	Serbia	flowering phase: undecane (6.6%), dodecanal (10.8%), germacrene D (14.1%); fruitforming phase: α-pinene (7.3%), β-pinene (26.1%), (Z)-β-ocimene (8.5%), (E)-ocimene (10.2%), bicyclogermacrene (7.7%), germacrene D (15.1%)	[44]
*H. salsugineum* N.Robson & Hub.-Mor.	Turkey	nonacosane (42.7%), hexadecanoic acid (23.2%), baeckeol (6.1%)	[27]
*H. scabroides* N.Robson & Poulter	Turkey	hexadecanoic acid (17.7%), spathulenol (5.3%), nonacosane (4.4%), dodecanoic acid (4.1%), baeckeol (4.1%), γ-muurolene (3.9%)	[27]
*H. scabrum* L.	Iran	α-pinene (50.0%), β-pinene (9.7%), limonene (6.6%), (E)-β-ocimene (5.6%), carvacrol (5.8%)	[57]
	Iran	α-pinene (40.9%), spathulenol (7.9%), β-pinene (5%), α-cadinol (4.7%), limonene (4.3%), epi-α-muurolol (3.2%)	[68]
	Iran	α-pinene (32.2%), β-pinene (9.2%), germacrene-D (7.1%), nonane (6.9%), limonene (6.4%), δ-cadinene (5.4%), 2-methyl-octane (3.8%), valencene (3.3%), 2-methyl-decane (3.3%), α-amorphene (3.10%), β-caryophyllene (2.1%)	[23]
	Turkey	α-pinene (74%), β-pinene (4.8%), myrcene (3.4%)	[69]
	Iran	α-pinene (70.2%), p-mentha-1,5-dien-8-ol (2.9%)	[64]
	Turkey	roots: undecane (66.1%); aerial parts: α-pinene (17.5%), γ-terpinene (17.4%), α-thujene (16.9%); flowers: α-pinene (55.6%), α-thujene (10.9%),γ-terpinene (7.7%); fruits oils: α-pinene (85.2%)	[70]
	Lebanon	α-pinene (37.8%), limonene (11.6%), myrcene (5.6%), β-pinene (3.4%), nonane (3%)	[71]
*H. silenoides* Juss.	Mexico	n-nonane (31.9%), α-pinene (16.1%), n-decanal (15.2%), 1-tridecanol (11.6%), n-dodecanal (10.5%)	[28]
*H. thymopsis* Boiss.	Turkey	α-pinene (44.0%), baeckeol (32.9%), spathulenol (8.0%), limonene (7.6%), camphene (5.2%)	[27]
*H. tomentosum* L.	Tunisia	α-pinene (3.7 or 26.3%), β-selinene (1.5 or 4.2%), n-pentacosane (57.0 or 0.6%), 1-heneicosene (10.3% or not detected), n-undecane (3.8 or 6.8%) (from different populations)	[60]
*H. triquetrifolium* Turra	Turkey	1-hexanal (18.8%), 3-methylnonane (12.5%), α-pinene (12.3%)	[29]
	Iran	germacrene-D (21.7%), β-caryophyllene (18.3%), δ-cadinene (6.4%), trans-β-farnesene (4.3%), α-humulene (3.8%), β-selinene (3.7%), γ-cadinene (3.3%), trans-phytol (3.2%)	[50]
	Greece	(E)-β-caryophyllene (27.9%), caryophyllene oxide (15.7%)	[31]
	Greece	α-pinene (13.9%), 3-methyl-nonane (10.2%), E-caryophyllenne (14.0%), caryophyllene oxide (9.7%), germacrene D (8.2%)	[22]
*H. umbellatum* A. Kern	Serbia	germacrene D (6.1%), (E)-nerolidol (4.4%), n-nonane (4.0%), (E)-caryophyllene (3.0%), caryophyllene oxide (3.0%)	[41]
*H. uniglandulosum* Hausskn. ex Bornm.	Turkey	2,6-dimethyl-3,5-heptadien-2-one (40.7%), nonacosane (3.2%), hexadecanoic acid (2.7%), α-pinene (2.7%)	[27]
	Turkey	α-pinene (35.1%), undecane (19.2%), benzoic acid (2.7%), cyclohexasiloxane (2.3%)	[51]

Plant botanical authorities according to IPNI.

During the first studies of *Hypericum* EOs, it was suggested that this group of compounds could serve as chemotaxonomic markers of the genus, and there are a few reports in the literature using statistical techniques to evaluate potential relations [44,45,46,47,48]. However, Crockett [17] pointed out the variability of the EOs at this taxonomic level and that such a hypothesis is not supported when examining data from a wide range of geographic distributions, taxonomic ranks, and seasonality. In general, the various parameters affecting the content, composition, and yields of *Hypericum* EOs could be related to the effect of variables such as genetic factors, developmental stages, and seasonal variation phenological cycle, types of plant material and specific organs used, methods of extraction, environmental conditions, and geographic distribution.

The extensive qualitative and quantitative variability reported for the chemical profile of the EOs from *H. perforatum* [17,19,20] was further supported by the studies of the last decade. The typical compounds being yielded in high amounts include monoterpenes such as α- and β-pinenes, sesquiterpenes such as germacrene-D, ishwarane, cadinenes, β-caryophyllene, and caryophyllene oxide, as well as several hydrocarbons such as n-decane, n-nonane, undecane, and dodecane (Table 1). Much variability has also been reported between subspecies, which is justified by the wide range of morphological variations characterizing these taxa [17]. *H. perforatum* ssp. *veronense* oil from Croatia showed α-pinene and n-nonane as the major constituents [49], while the oil from Greece yielded high amounts of selinenes [31]. Some homogeneity is reported for the composition of the EO of *H. scabrum* from different collection sites, showing that α-pinene is the most represented compound (Table 1). Another study on hypocotyl explants of ten different wild populations of *H. scabrum* evaluated the callus essential oils production with industrial application. According to the analyses, a total of forty-one components were detected with relatively high variation in their essential oil composition. Among constituents, α-pinene (7.6–40.2%), β-pinene (1.3–35.7%), limonene (0.0–32.2%), β-ocimene (0.0–37.9%), and germacrene D (0.2–30.6%) were found as the most abundant constituents [26]. Moreover, significant variability has been documented for the EOs of *H. triquertifolium*, which, when collected in Greece, yielded either high amounts of α-pinene together with β-caryophyllene [22] or high percentages of β-caryophyllene and very low quantities of α-pinene [31]. Much diversity has been reported for this plant when collected from different populations in Tunisia [45], while the oils from Iran and Turkey were abundant in germacrene-D and hexanal, respectively [29,50]. Tahir et al. [25] characterized the EO from *H. triquetrifolium* cultures produced by the root and stem for the first time and cited that alkane, aldehyde, and monoterpene compounds are the foremost fractions. Furthermore, *H. thymopsis* and *H. scabroides* showed different results when collected from other locations in Turkey [27]. *H. uniglandulosum* collected again in Turkey presented α-pinene as the major compound of the EO (35.1%) [51], though Ahmed et al. [27] reported much lower levels of this compound (2.7%), while 2,6-dimethyl-3,5-heptadien-2-one was found to be the major constituent (40.7%). Similar differences in the EOs composition of *H. lydium* collected from Turkey were observed, reporting, on the one hand, α-pinene (58%; 71.2%) [51,52] as the major component, and on the other hand, oxygenated terpenes (verbenone 22.2%, caryophyllene oxide 18.3% [53]). In the latter study, *H. orientale* also collected from Turkey showed high levels of β-selinene (37.1%), though samples collected from France were abundant in α-pinene and undecane [53]. The major compounds for *H. confertum* (syn. *H. saturejifolium*) from different collection sites in Turkey were identified as germacrene D (30.2%) [42], α-pinene (7.8%), γ-muurolene (7.2%), or δ-cadinene (6.5%) [53]. The main compound of *H. hircinum* EO from Greece was (E)-caryophyllene (65.9%), while samples collected in Italy identified nonane or cis-guaiene as the major compounds [21]. The EO from H. hircinum ssp. majus also afforded cis-guaiene as the main metabolite [54]. Several studies reported α-pinene to be the main constituent of EOs from *H. empetrifolium* collected from Greece [21,39], while when the plant material was collected in Turkey, a sesquiterpene hydrocarbon, i.e., alloaromadendrene, was among the top major compounds [55]. Another sesquiterpene hydrocarbon was identified as the main component of *H. empetrifolium* ssp. empetrifolium collected in Greece, (E)-β-farnesene (29.5%) [31]. *H. capitatum* similarly presented differences in the composition of its EO when collected from different locations, more specifically, spathulenol (12.9%) and iso-longifolene (11.2%) or α-pinene (20.3%) were reported as the main constituents from different collection sites in Turkey [56]. Likewise, *H. helianthemoides* from different collection sites in Iran showed β-caryophyllene as the major metabolite [20], which, in a later study, was found in lower levels and the main constituent was α-pinene [57]. H. patulum from China showed nonane as the major constituent [58], while an older study presented α-pinene as the main compound, which was also at high levels in a sample from Iran, although the rest of the composition of these EOs was different, for example, β-pinene, which was the main constituent in the EO from Iran was absent in the oil from China [59]. A recent study on *H. pseudohenryi* showed different results in comparison to a previous study and reports the occurrence of highly polar compounds, which is justified by the use of supercritical CO_2_ extraction [24]. Regarding *H. perfoliatum*, a recent study showed that the two most abundant constituents in the EO from Greece (γ-muurolene and δ-cadinene) were described at much lower levels in previous studies [31]. In addition to the documentation of the EO content of inflorescences and leaves from *Hypericum* plants, the aroma of the berry-like fruits from *H. androsamemum* was chemically investigated, showing mainly monoterpene hydrocarbons and especially limonene as the most abundant compound [43].

### 3.2. Bioactivities from Hypericum Essential Oils

During the last decade, further studies were performed using the EOs from *Hypericum* plants and evaluating their biological effects. In fact, 23 of the 27 plants in Table 2 are described herein for new biological effects in comparison with former surveys [20]. Previously, many reports on antimicrobial activities, only a few on insecticidal effects [20], and two studies evaluating antiangiogenetic (*H. perforatum*, using the chicken chorio allantoic membrane assay), and antioxidant (*H. undulatum*) activities were published [20]. Out of the studies presented in Table 2, 17 reports include antimicrobial studies (antibacterial and antifungal) together with an evaluation of other activities, while 10 reports are inclusive of antimicrobial effects. As there is a variety of assays used for the evaluation of antimicrobial potential from *Hypericum* EOs (such as disc diffusion, microdilution, and measurement of MICs assays), it is not easy to make a direct comparison between the observed efficacies between different studies. Moreover, the main constituents characterizing several *Hypericum* EOs, such as α-pinene, β-pinene, and (E)-caryophyllene, also possess significant antimicrobial activities, which justifies the exploration of the genus EOs as antimicrobial agents [20]. In addition, further biological effects of *Hypericum* EOs are gathered (in vitro: Antiviral, antimalarial, cytotoxic, neuroprotective, tyrosinase inhibitory, and immunomodulatory activities; in vivo: Anti-angiogenic, hepatoprotective, and wound-healing effects), attempting to bridge the potential relationship between traditional uses and *Hypericum* EOs’ biological effects.

**Table 2 molecules-27-05246-t002:** Survey of the biological effects exerted from *Hypericum* spp. EOs.

*Hypericum* spp.	Plant Origin	EO Biological Activities	Reference
*H. aegypticum* ssp. *webbii* (Spach) N. Robson	Greece	antibacterial activity (*Bacillus subtilis*, *Enterococcus faecalis*, *Escherichia coli*, *Klebsiella pneymoniae*, *Micrococcus luteus*, *Pseudomonas aeruginosa, Salmonella abony, Staphylococcus aureus*, *S. epidermis*); antifungal activity *(Candida albicans*)	[38]
*H. amblyocalyx* Coustur. & Gand	Greece	antibacterial activity (*Aspergillus fumigatus*, *Bacillus cereus, Escherichia coli*, *Listeria monocytogenes*, *Pseudomonas aeruginosa, Staphylococcus aureus*); antifungal activity (*Candida tropicalis, Candida krusei*, *Penicillium funiculosum*, *Penicillium verucosum*)	[39]
*H. annulatum* Moris	Serbia	antibacterial activity (*Bacillus subtilis, Escherichia coli*, *Pseudomonas Aeruginosa*, *Salmonella abony, Staphylococcus aureus*); antifungal activity (*Aspergillus niger*, *Candida albicans*)	[41]
*H. bellum* H.L.Li	China	neurite outgrowth-promoting assay; neuroprotective activity assay; antibacterial activity (*Escherichia coli*, *Pseudomonas aeruginosa*, *Salmonella enterica* ssp. *enterica Staphylococcus aureus* ssp. *aureus*), antifungal activity (*Candida albicans*); tyrosinase inhibitory assay	[24]
*H. canariense* L.	Canary Islands	antiproliferative (A375 and MDA-MB 231, HCT116 cells by MTT assay); antioxidant activity (phenolic content, DPPH, ABTS and FRAP assays); antibacterial activity (*Staphylococcus aureus*, *Escherichia coli*, *Pseudomonas aeruginosa*, *Enterococcus faecalis*; antifungal activity (*Candida albicans*)	[32]
*H. saturejifolium* Jaub. & Spach (syn. *H. confertum* Choisy)	Turkey	anti-angiogenic effects using the chick embryo chorioallantoic membrane (CAM) assay	[42]
*H. elegans* Steph. ex Willd.	Serbia	antibacterial activity (*Bacillus subtilis, Escherichia coli*, *Pseudomonas aeruginosa*, *Staphylococcus aureus*, *Salmonella abony*); antifungal activity (*Aspergillus niger*, *Candida albicans*)	[41]
*H. empetrifolium* Willd.	Greece	antibacterial activity (*Bacillus cereus, Escherichia coli*, *Listeria monocytogenes*, *Pseudomonas aeruginosa*, *Staphylococcus aureus)*; antifungal activity (*Aspergillus fumigatus*, *Penicillium funiculosum*, *P. verucosum*, *Candida albicans*, *C. tropicalis*, *C. krusei*)	[39]
	Greece	wound healing in vivo using SKH-hr1 mice	[22]
*H. gaitii* Haines	India	antioxidant activity (DPPH, ABTS, reducing power assay)	[34]
*H. grandifolium* Choisy	Canary Islands	antiproliferative (A375 and MDA-MB 231, HCT116 cells by MTT assay); antioxidant activity (phenolic content, DPPH, ABTS and FRAP assays); antibacterial activity (*Staphylococcus aureus*, *Escherichia coli*, *Pseudomonas aeruginosa*, *Enterococcus faecalis*; antifungal activity (*Candida albicans*)	[32]
*H. helianthemoides* (Spach) Boiss.	Iran	antibacterial (*Bacillus cereus*, *Listeria monocytogenes*, *Proteus vulgaris*, *Salmonella typhimurium*); antioxidant activity (DPPH)	[57]
*H. ascyron* L. (syn. *H. hemsleyanum* H.Lév. & Vaniot)	China	insecticidal activity (repellency of three plant essential oils against red flour beetle *Tribolium castaneum*)	[40]
*H. hircinum* L.	Turkey	anti-angiogenic effects using the chick embryo chorioallantoic membrane (CAM) assay	[42]
*H. hircinum* L. ssp. *majus* (Aiton) N. Robson	Italy	antioxidant activity (DPPH, ABTS); antiproliferative activity (human glioblastoma (T98G), human prostatic adenocarcinoma (PC3), human squamous carcinoma (A431) and mouse melanoma (B16-F1) tumor cell lines by MTT assay)	[54]
*H. hookerianum* Wight & Arnott	China	neurite outgrowth-promoting assay; neuroprotective activity assay, antibacterial activity (*Escherichia coli*, *Pseudomonas aeruginosa*, *Salmonella enterica* ssp*. enterica*, *Staphylococcus aureus ssp. aureus*,); antifungal activity (*Candida albicans*); tyrosinase inhibitory assay	[24]
*H. humifusum* L.	Tunisia	insecticidal (larvicidal) activity (*Culex pipiens*)	[60]
*H. jovis* Greuter	Greece	antibacterial activity (*Bacillus cereus*, *Escherichia coli*, *Listeria monocytogenes*, *Pseudomonas aeruginosa*, *Staphylococcus aureus)*; *antifungal activity (Aspergillus fumigatus, Penicillium funiculosum*, *P. verucosum*, *Candida tropicalis*, *C. krusei*)	[39]
*H. lydium* Boiss.	Turkey	antioxidant activity (on liposome peroxidation, DPPH, superoxide radical scavenging activity, non-site and site-specific hydroxyl radical-mediated 2-deoxy-d-ribose degradation)	[52]
*H. maculatum* Crantz	Serbia	antibacterial activity (*Bacillus subtilis*, *Escherichia coli*, *Pseudomonas aeruginosa*, *Salmonella abony*, *Staphylococcus aureus*); antifungal activity (*Aspergillus niger*, *Candida albicans*)	[46]
*H. microcalycinum* Boiss. & Heldr. [syn. *H. hyssopifolium* Chaix ssp. *elongatum* (Ledeb.) Woron var. *microcalycinum.* (Boiss. & Heldr.) Boiss.]	Turkey	anti-angiogenic effects using the chick embryo chorioallantoic membrane (CAM) assay	[42]
*H. patulum* Thumb.	China	antioxidant activity (DPPH and ABTS^+^ radicals scavenging assays	[58]
*H. perforatum* L.	Romania	antibacterial activity (*Enterococcus faecalis*, *Escherichia coli*, *Klebsiella pneumoniae*, *Pseudomonas aeruginosa*, *Salmonella typhimurium*, *Staphylococcus aureus*); antifungal activity (*Candida albicans*)	[62]
	Iran	insecticidal effects against *Tribolium castaneum*	[63]
	Tunisia	larvicidal activity (*Culex pipiens*)	[60]
	Serbia	antifungal activity (*Candida albicans*)	[86]
	Turkey	insectisidal against adults of Colorado potato beetle, *Leptinotarsa decemlineata*	[79]
	Turkey	insecticidal activity (fumigant Toxicity against *Sitophilus zeamais)*	[78]
	Iran	antioxidant activity (β-carotene bleaching and DPPH); antibacterial (*Escherichia coli*, *Staphylococcus aureus*)	[64]
	Iran	antibacterial (*Bacillus cereus*, *Listeria monocytogenes*, *Proteus vulgaris*, *Salmonella typhimurium*); antioxidant activity (DPPH)	[57]
	Albania	antioxidant activity (Inhibition of linoleic acid lipid peroxidation, soybean lipoxygenase inhibition, DPPH)	[65]
	Albania	antimicrobial activity (*Escherichia coli*, *Enterococcus faecalis*, *Klebsiella pneumoniae Pseudomonas aeruginosa*, *Salmonela typhimurium*, *Staphylococcus aureus*); antifungal activity (*Candida albicans*)	[66]
	Turkey	insecticidal activity on *Sitophilus granarius*	[77]
	Turkey	anti-angiogenic effects using the chick embryo chorioallantoic membrane assay	[42]
	Greece	wound healing in vivo using SKH-hr1 mice	[22]
	China	neurite outgrowth-promoting assay; neuroprotective activity assay; antibacterial activity (*Escherichia coli*, *Pseudomonas aeruginosa*, *Salmonella enterica* ssp. *Enterica*, *Staphylococcus aureus* ssp. *aureus*); antifungal activity (*Candida albicans*); tyrosinase inhibitory assay	[24]
	USA	immunomodulatory activity	[67]
*H. perforatum* L. ssp. *veronense* (Schrank) H. Lindb.	Croatia	antiproliferative (HeLa, HCT116, U2OS); antioxidant activity (ORAC, DPPH); antiphytoviral (Tobacco mosaic virus) activities	[49]
*H. pseudohenryi* N.Robson	China	neurite outgrowth-promoting assay; neuroprotective activity assay; antibacterial activity (*Escherichia coli*, *Pseudomonas aeruginosa*, *Salmonella enterica* ssp. *enterica*, *Staphylococcus aureus* ssp. *aureus);* antifungal activity (*Candida albicans*); tyrosinase inhibitory assay	[24]
*H. reflexum* L.	Canary Islands	antiproliferative (A375 and MDA-MB 231, HCT116 cells by MTT assay); antioxidant activity (phenolic content, DPPH, ABTS and FRAP assays); antibacterial activity (*Staphylococcus aureus*, *Escherichia coli*, *Pseudomonas aeruginosa*, *Enterococcus faecalis*; antifungal activity (*Candida albicans*)	[32]
*H. rochelii* Griseb. & Schenk	Serbia	antibacterial activity (*Bacillus subtilis*, *Escherichia coli*, *Pseudomonas aeruginosa*, *Salmonella abony*, *Staphylococcus aureus*); antifungal activity (*Aspergillus niger*, *Candida albicans*)	[41]
*H. scabrum* L.	Turkey	biting deterrent activity against *Aedes aegypti*; antimalarial activity against *Plasmodium falciparum*; *Mycobacterium intracellulare*; antifungal activity (*Cryptococcus neoformans*, *Candida krusei*)	[69]
	Turkey	insecticidal effects against adults of *Leptinotarsa decemlineata* Say	[79]
	Iran	antioxidant activity (β-carotene bleaching and DPPH); antibacterial (*Escherichia coli*, *Staphylococcus aureus*)	[64]
	Iran	modulating effect on hepatic metabolizing enzymes in vivo in rats treated by acetaminophen	[83]
	Iran	hepatoprotective effects against oxidative stress induced by acetaminophen in vivo in rats	[68]
	Iran	antioxidant activity (DPPH and β-carotene assays)	[82]
	Iran	antibacterial activity (*Bacillus cereus*, *Listeria monocytogenes*, *Proteus vulgaris*, *Salmonella typhimurium*); antioxidant activity (DPPH)	[57]
	Turkey	insecticidal activity (*Sitophilus granarius)*	[77]
	Turkey	antibacterial activity (*Escherichia coli*, *Staphylococcus aureus*, *Bacillus subtilis*); antifungal activity (*Candida albicans*, *C. tropicalis*); antioxidant activity (DPPH)	[70]
	Lebanon	antibacterial activity (*Pseudomonas aeruginosa, Staphylococcus aureus*); *antifungal activity* (*Candida albicans*, *Trichophyton rubrum*, *T. mentagrophytes*, *T. soudanense, T. violaceum*, *T. tonsurans*); synergistic effect with amphotericin B	[71]
*H. tomentosum* L.	Tunisia	insecticidal (larvicidal) activity (*Culex pipiens*)	[60]
*H. triquetrifolium* Turra	Tunisia	antibacterial activity (*Aeromonas hydrophila*, *Bacillus cereus*, *Enterococcus faecalis*, *Escherichia coli*, *Pseudomonas aureginosa*, *Salmonella typhimurium*, *Staphylococcus aureus*, *Staphylococcus epidermidis, Vibrio cholerae*); antifungal activity (*Aspergillus niger*, *Fusarium solani*, *Botrytis cinerea*, *Candida albicans*, *Candida glabrata*, *Candida krusei*); antiviral activity (Coxsakievirus) activities	[72]
	Greece	wound healing in vivo using SKH-hr1 mice	[22]
*H. umbellatum* A. Kern	Serbia	antibacterial activity (*Bacillus subtilis, Escherichia coli*, *Pseudomonas aeruginosa*, *Salmonella abony, Staphylococcus aureus*); antifungal activity (*Aspergillus niger*, *Candida albicans*)	[41]

Plant botanical authorities according to IPNI.

#### 3.2.1. In Vitro Studies

##### Antibacterial Activity

Gram Positive bacteria: The antimicrobial evaluation against *Micrococcus luteus* (ATCC 9341) and *Staphylococcus epidermidis* (ATCC 12228) showed moderate effects from the EOs of *H. aegypticum* ssp. *webbii* [38], although previous studies from other plants reported average to good MIC values against these bacteria [20]. Similarly, *S. epidermidis* was reported mostly as resistant to the EOs from *H. triquetrifolium* plants from different collection points [72]. In the same study, the EOs from *H. triquertifolium* showed, in general, a wide range from moderate to potent activity against other Gram-positive bacteria (such as *Enterococcus faecalis*, *Staphylococcus aureus*, *Bacillus cereus*), which could be supported by the chemical variation due to environmental parameters. Within the evaluated *Hypericum* EOs, *S. aureus* is on the front line of antibacterial screening, being included in twenty-four studies of sixteen *Hypericum* spp. (Table 2). Moderate effects were presented for *H. aegypticum* ssp. *webbii* and *H. perforatum* EOs against *Enterococcus faecalis* [38] and similar results were reported for *H. perforatum* using the disk diffusion assay [62,66]. Five Hypericum species showed good inhibitory activity against *Listeria monotocytogene* in two studies. More specifically, the MIC values were 0.020 mg/mL and 0.010 mg/mL for *H. jovis* and *H. amblyocalyx*, respectively, which were comparable with the MIC values observed for the positive controls streptomycin (MIC 0.2 mg/mL) and ampicilin (MIC 0.4 mg/mL) [39]. Likewise, the antimicrobial effects against *L. monocytogene* were evaluated for the EOs from *H. helianthemoides* (MIC 125 μg/mL), *H. scabrum* (MIC 62 μg/mL), and *H. perforatum* (MIC 250 μg/mL), with flumequine (MIC 125 μg/mL), ciprofloxacin (MIC 62 μg/mL), and ampicillin (MIC 62 μg/mL) used as positive controls [57]. The two latter studies also tested the same EOs against *Bacillus cereus,* where good antibacterial activity was observed for *H. empetrifolium* [39] and *H. scabrum* [57], which, in comparison with the other under-investigation species, included higher levels of α-pinene. Furthermore, *H. scarbum* presented selective activity with IC_50_ value of 52.98 μg/mL against *Mycobacterium intracellulare* (ATCC 23068), a Gram-positive bacterium that was not tested before [69].

Gram Negative bacteria: Pirbalouti et al. [57] mentioned the antimicrobial evaluation against *Proteus vulgaris*; however, by oversite, the respective table provided data against *Pseudomonas aureginosa* (showing moderate-to-good inhibitory activities from *H. helianthemoides*, *H. scabrum,* and *H. perforatum*). In the same study, the antimicrobial effects against *Salmonella typhimurium* were evaluated for the EOs from *H. helianthemoides* (MIC 125 μg/mL), *H. scabrum* (MIC 125 μg/mL), and *H. perforatum* (MIC 500 μg/mL), with the controls flumequine (MIC 62 μg/mL), ciprofloxacin (MIC 125 μg/mL), and ampicillin (MIC 125 μg/mL) [57]. *Salmonella typhimurium* strain ATCC 14028 was sensitive against *H. triquertifolium* EOs from different populations with a range of MIC values from 0.39 to 25.00 mg/mL [72]. Moleriu et al. [62] mentioned strong antimicrobial activity from the EO of *H. perforatum* against the same strain using the disc diffusion method. Several EOs from *Hypericum* species showed weak to moderate effects against *Salmonella abony* NCTC 6017 [38,41,46]. Another *Salmonella* sp. (*S. enterica* ssp. *enterica* ATCC14028) was tested using the EOs from *H. perforatum*, *H. hookerarium,* and *H. belum*, and the inhibition rate varied from 1.2 to 32% [24]. Rouis et al. [72] tested Vibrio cholera (ATCC 39315) and *Aeromonas hydrophila* (ATCC 7966) for the first time, and *H. triquertifolium* EO showed bacteriostatic effects; however, the results could not be correlated with the main constituents of the EOs from the different collection sites. *H. perforatum* was found to be active against *Klebsiella pneumoniae* (ATCC 13882) using the disc diffusion assay [62,66], while for the strain NCIMB 9111, both *H. perforatum* and *H. aegypticum* ssp. *webbii* showed moderate activity using the broth microdilution method [38]. Contradictory results have been reported for *Escherichia coli* and *Pseudomonas aureginosa*, being either sensitive or resistant against the EOs from *Hypericum* spp. [24,32,39,41,64,70]. The observed differences could be attributed to experimental parameters such as the selected protocols, the tested concentrations, or the controls, as well as the chemical profile of the samples.

##### Antifungal Activity

In the last decade, many *Hypericum* species were used for the antifungal evaluation of their EOs (Table 2). Specifically, similar potent effects were reported against three fungi from the *H. triquertifolium* EOs collected in several sites in Tunisia, showing MIC and MFC values of 3.12 μg/mL for *Botrytis cinerea* (tested for the first time in genus EO), Fusarium solani, and *Aspergilus niger* [72]. In the same study, the best antifungal activity was exerted against *Candida glabrata,* and the MIC and MFC values were 0.39 μg/mL and 1.56 μg/mL, respectively. The EO from H. scabrum showed selective antimicrobial activity against *Cryptococcus neoformans* (ATCC 90113) (tested for the first time in genus EO) (IC_50_ 34.71 μg/mL), while the IC_50_ for amphotericin B was 0.32 μg/mL [69]. Two further species tested for the first time, *Aspergillus fumigatus* and *Penicillium funiculosum*, were sensitive against the EOs from *H. jovis* (MIC and MFC values of 0.015 and 0.030 mg/mL), *H. empetrofolium* (MIC and MFC values of 0.030 and 0.060 mg/mL), and *H. amblyocalyx* (MIC and MFC values of 0.010 and 0.020 mg/mL) [39]. Moreover, *H. jovis* (MIC and MFC values of 0.025 and 0.050 mg/mL), *H. empetrifolium* (MIC and MFC values of 0.030 and 0.060 mg/mL), and *H. amblyocalyx* (MIC and MFC values of 0.010 and 0.020 mg/ml) were active against *Penicillium verucosum* (also evaluated for the first time); the yeasts *C. tropicalis* and *C. krusei* were more sensitive, with MIC values of 0.001–0.010 mg/mL and MFC values of 0.002–0.030 mg/mL, while no remarkable activity was detected against *A. niger* and *C. albicans*. *H. empetrifolium*, which presented greater activity, yielded higher amounts of monoterpene hydrocarbons (especially α-pinene) in comparison with the other under-investigation species [39]. *H. scabrum* EO obtained from flowers was found active against the same strain of *C. tropicalis* (ATCC 750) with an MIC value of 312.5 μg/mL, while the MIC value from the EO from aerial parts was 156.25 μg/mL, which could be explained by the differences in the levels of α-pinene in the biomass used (55.6% in flowers vs. 17.5% in aerial parts) [70]. Antimicrobial effects of *Hypericum* EOs against *C. krusei* were firstly evaluated during the last decade, for a clinical isolate as mentioned above [39], as well as the strain ATCC 6258 for *H. triquertifolium* EOs [72] and *H. scabrum* [69], which showed selective antimicrobial activity with IC_50_ 104.43 μg/mL (amphotericin B used as positive control showed IC_50_ 0.52 μg/mL). Several studies reported the screening against *Candida albicansi*, featuring mainly fungal resistance or low effects from Hypericum EOs. Moreover, *Aspergillus niger* appeared to be the most resistant in several studies [39,41,46], while Rouis et al. [72] reported weak to moderate effects for *H. triquertifolium* EOs.

##### Antiviral Activity

EOs have been screened against several pathogenic viruses, and their components may act synergistically or potentiate other antiviral agents, or even provide symptom relief [73,74]. *H. triquertifolium* EOs did not show antiviral activity against the coxsakie virus B3 Nancy strain, whether incubated with the virus prior to infection or incubated with Vero cells before the inoculation [72]. However, the activities against other viruses cannot be ruled out because of findings regarding antiviral activities of either EOs from other plants [74] or from *Hypericum* extracts [75]. Moreover, the recent study by Vuko et al. [49] mentioned that the EO from *H. perforatum* was an effective antiphytoviral agent against Tobacco mosaic virus.

##### Antimalarial Activity

*Hypericum* has been traditionally used for the treatment of malaria, and there are a few studies investigating the antimalarial activity of its extracts. These studies showed promising results for some extracts and isolated compounds, while the first report for the investigation of the EO from *H. scabrum* showed weak antimalarial activity against two strains of *Plasmodium falciparum* with IC_50_ values of 28.8 μg/mL (for D6) and 15.7 μg/mL (for W2). Moreover, both strains appeared to be resistant to the pure compounds (α- and β-pinenes and myrcene) [69].

##### Insecticidal Activity

The insecticidal activity of many plant materials, such as extracts and EOs, has been evaluated against different pests, and monoterpenoids are regarded as the responsible constituents for the observed effects [76]. In the current review, the EOs obtained from *Hypericum* spp. showed insecticidal effects against pests found in stored products and crops, as well as against mosquitos. Parchin et al. [63] reported a remarkable biological response from *H. perforatum* EO by increasing the mortality rate and acting as an antifeedant agent against the adults of the red flour beetle (*Tribolium castaneum* Herbst.), which is a major pest in stored products. The EO from *H. hemsleyanum* was tested against the same pest at a concentration of 31.5 μg/cm^2^ and showed the strongest repellency throughout the experiment (72 h) [40]. *H. perforatum* and *H. scabrum* have been shown to possess insecticidal activities against two further pests related to stored products, i.e., *Sitophilus granarius* [77] and *S. zeamais* [78]. Another study included the EOs from *H. perforatum* and *H. scabrum* among other tested medicinal plants and showed low insecticidal effects against the Colorado potato beetle (*Leptinotarsa decemlineata*) in comparison with the other taxa [79]. *H. scabrum* EO was tested using an in vitro mosquito-biting bioassay against female *Aedes aegypt* and showed higher biting deterrent activity than the solvent control, but the activity was significantly lower than the positive control, DEET (N,N-diethyl-3-methylbenzamide) [69]. EOs from *H. tomentosum*, *H. humifusum,* and *H. perforatum* were subjected to a larval toxicity assay against another mosquito, i.e., *Culex pipiens larvae* [60]. All four oils possessed larvicidal properties, with the EO from *H. tomentosum* being the most promising.

##### Cytotoxic Activity

In the last decade, the first attempts to evaluate the antiproliferative effects of *Hypericum* EOs were made by Zorzetto et al. [32] reporting strong cytotoxic activity against three human cell lines (A375 human malignant melanoma, MDA-MB 231 human breast adenocarcinoma, and HCT116 human colon carcinoma), using the MTT assay. The authors suggested that synergistic effects take place in the observed cytotoxicity, as previous studies on the commercially available major constituents such as α-pinene, β-pinene, (E)-caryophyllene, and n-nonane exhibited lower activities than the EOs. However, low cytotoxic effects were observed from *H. triquertifolium* EOs using an animal cell line (Vero cells from *Chlorocebus* sp. kidney) [72], as well as from *H. hircinum* ssp. *majus* EO against three human and one animal cell lines (human glioblastoma (T98G), human prostatic adenocarcinoma (PC3), human squamous carcinoma (A431), and mouse melanoma (B16-F1)) using the MTT assay [54]. In a recent study, *H. perforatum* ssp. *veronense* EO showed moderate results against three human cell lines (cervical cancer HeLa, human colon cancer HCT116, and human osteosarcoma U2OS) using the MTS-based cell proliferation assay [49].

##### Antioxidant Activity

Many studies have been conducted on the antioxidant capacity of *Hypericum* species featuring polar extracts and rarely essential oils. Based on our survey, EOs from eleven species have been investigated so far for their antioxidant effects by in vitro methods such as the evaluation of total phenolic content or β-carotene, DPPH, FRAP, ABTS, and ORAC assays, which have attributed their potential to phenolic compounds (terpenoid and phenylpropanoid), which accounted for some of the principal components of the EOs. The antioxidant activities of *H. scabrum* and *H. perforatum* EOs were evaluated using the β-carotene/linoleic acid and DPPH assays, and the higher activities observed on the DPPH assay were attributed to the synergy of compounds, although the effects were proportional to the levels of α-pinene [64]. The EOs from *H. helianthemoides*, *H. perforatum,* and *H. scabrum* were tested using the DPPH assay, and the authors suggested that the highest observed antioxidant effects of *H. scabrum* could be correlated to the relatively high amounts of α-pinene [57]. Zorzeto et al. [32] investigated the antioxidant power of three species, *H. grandifolium*, *H. reflexum,* and *H. canariense*, by ABTS, DPPH, and FRAP assays resulting in the most active being *H. grandifolium* (from La Esperanza); however, the activities were low in comparison to the respective methanol-acetone extracts. In this case, the authors concluded that the higher levels of oxygenated sesquiterpenes positively contribute to the total antioxidant effects of the EOs. A recent study showed higher antioxidant activity of *H. perforatum* ssp. *veronense* EO than hydrosol in both ORAC and DPPH methods [49]. Another study on *H. gaitii* EO showed moderate antioxidant capacity using the DPPH and ABTS assays [34]. On the other hand, relatively high antioxidant effects were found using DPPH and ABTS radical scavenging assays on *H. hircinum* ssp. *majus* EO with IC_50_ 680 and 270 mg/mL, respectively (vs. Trolox 32.5 mg/mL and ABTS 24.4 mg/mL), and the main compound was cis-β-guaiene [54]. Similar results were reported for *H. scabrum* EO (major constituent α-pinene 40.9%), which had enough strong radical scavenging activity using the DPPH assay with 18.5% inhibition in comparison to Trolox 36.1%, as well in the β-carotene assay showing 59.9% inhibition in the formation of peroxidation products compared to BHT (91.7%) [68]. The antioxidant potential of EOs has been studied in correlation to their chemistry, and autoxidation occurs in the case of α-pinene or similar components [80].

##### Neuroprotective Activity

In a recent study, the EOs from *H. perforatum*, *H. hookerianum*, *H. bellum,* and *H. pseudohenryi* were evaluated in the neurite outgrowth-promoting and corticosterone-induced neurotoxicity assays using PC12 cells [24]. *H. bellum* demonstrated the most significant neurite-promoting and neuroprotective activity, and had high levels of sesquiterpenes and especially curdione (30.9%), which presented neuroprotective effects in vivo [81].

##### Tyrosinase Inhibitory Activity

*H. perforatum*, *H. hookerianum*, *H. bellum*, as well as *H. pseudohenryi* are traditionally used in China for skin care, thus a recent study by Ji et al. [24] evaluated their EOs in the tyrosinase inhibitory assay. Only the EOs from *H. perforatum* (collected in Wushan) and *H. pseudohenryi* showed tyrosinase inhibition activity but were less active than that of the positive control, kojic acid. Better results on tyrosinase inhibition have been obtained with aqueous or methanolic extracts of *Hypericum* plants [24].

##### Immunomodulatory Activity

Recently, *H. perforatum* EOs were evaluated for their immunotherapeutic properties by applying Ca^2+^ mobilization, chemotaxis, reactive oxygen species (ROS) production, kinase Kd determination, and elastase inhibition assays using human neutrophils [67]. The EO from the leaves was more active in the inhibition of neutrophil Ca^2+^ mobilization, chemotaxis, and ROS production, and yielded higher amounts of mono- and sesqui- terpene hydrocarbons in comparison to the EO from the flowers. The major compounds, such as germacrene D, β-caryophyllene, and α-humulene, were suggested as the active compounds of the EOs from the leaves as they also inhibited the neutrophil responses [67].

#### 3.2.2. In Vivo

##### Anti-Angiogenic Activity

Excessive angiogenesis characterizes a variety of disorders such as cancer, autoimmune diseases, diabetic retinopathy, obesity, etc. The anti-angiogenic effects of four Hypericum EOs were evaluated in vivo using the chick embryo chorioallantoic membrane assay [42]. *H. perforatum* EO showed the most significant anti-angiogenic effects, whereas *H. confertum*, *H. hircinum*, and *H. hyssopifolium* ssp. *elongatum* var. *microcalycinum* showed no activity compared to the controls. Though α-pinene was the major metabolite in the EO from *H. perforatum*, other under-investigation species yielding even higher concentrations of this compound were inactive, which suggests the synergistic effects of the minor compounds in the EO of *H. perforatum* [42].

##### Hepatoprotective Activity

Acetaminophen (paracetamol) is widely used over the counter for pain relief, and acute overdoses are responsible for liver damage. The protective role of *H. scabrum* EO was evaluated on acetaminophen-induced liver damage using Wistar rats in two studies [82,83]. The authors reported that the hepatoprotective effects could be attributed to the modulation of the hepatotoxicity induced by the acetaminophen by adjusting the oxidative stress/antioxidant parameters [82] or rebalancing cytochrome P450, glutathione s-transferase (GST), and liver function markers [83]. The major constituent of *H. scabrum* EO was identified as α-pinene (40.9%) [68].

##### Wound Healing Activity

In a recent study, the wound healing efficacy of EOs from *Hypericum* spp. was evaluated and compared between species [22]. It is noteworthy that the content of the translucent glands of *Hypericum* plants (EOs and phloroglucinols) is extracted in the infused oil (Oleum Hyperici) [17], which is the well-known preparation, used since ancient Greek times for the treatment of wounds, thus justifying the evaluation of *Hypericum* EOs as wound healing agents. *H. empetrifolium* possessed the most significant healing properties while using *H. perforatum* and *H. triquetrifolium* EOs, but skin inflammation persisted in an in vivo model using hairless SKH-hr1 mice [22]. *H. empetrifolium* (whose EO yielded higher levels of α-pinene (19.0%, vs. 6.4% and 13.9% for *H. perforatum* and *H. triquetrifolium*, respectively)) shows long-term use in Greece for wounds and skin inflammations and it may be the ‘hypericon’ quoted by Dioscorides, since the previously reported *H. coris* does not grow wild in Greece [22].

### 3.3. Potential Relationship between Traditional Uses and Hypericum Essential Oils Activities

The EOs from *Hypericum* spp. showed promising results as wound healing agents in vivo [22], and since they could be important constituents of the infused oil (*Oleum Hyperici*), they are highly likely to contribute to the wound healing efficacy of *Hypericum* preparations. Moreover, several microorganisms, such as *Staphylococcus aureus*, *Pseudomonas aeruginosa*, and *Escherichia coli*, are frequently isolated from skin wounds in humans and animals [84], thus the well-documented antimicrobial activities from *Hypericum* EOs could play an important role in order to understand the wound healing effects of *Hypericum*. It is also noteworthy that the enzyme inhibitory activities of *Hypericum* EOs are based on their traditional use for skin care [24]. The studies so far suggest that *Hypericum* EOs have only weak activity against *Plasmodium falciparum*; consequently, other compound classes produced from *Hypericum* plants are responsible for their traditional use as antimalarial agents [85]. The observed hepatoprotective activity of *Hypericum* EOs in vivo [68,83] could correlate with several reports of traditional uses of *Hypericum* plants for liver problems [20]. Moreover, as oxidative stress and inflammatory pathways are linked to depression, the antioxidant effects from *Hypericum* EOs [54,57] could be involved in the anti-depressant efficacies of *Hypericum* preparations. Taking into account the long-term use of *Hypericum* in folk medicine, and in order to establish the traditional uses, further studies based on modern techniques should be conducted on this topic to fulfill the requirement for information on the efficacy and safety of *Hypericum* preparations.

## 4. Conclusions

Essential oils are highly concentrated complex mixtures biosynthesized by plants to serve specific biological functions, including endogenous defense mechanisms, interaction with other organisms, and adaptations to the environment. Every year, many research papers are produced on this topic, considering EOs’ highly valuable sources due to their aromatic and medicinal properties. In this framework, the present review covers literature data from various scientific databases for the period of 2012–2022 and summarizes the existing knowledge on the chemical composition of *Hypericum* EOs, their modern pharmacological data, and, in parallel, attempts to correlate up-to-date knowledge with its traditional uses. Chemically, *Hypericum* EOs include, among others, monoterpene hydrocarbons (α- and β-pinene), sesquiterpene hydrocarbons (E-caryophyllene and germacrene D), and oxygenated sesquiterpenes (spathulenol and caryophyllene oxide), while major constituents are, in some cases, n-alkanes (undecane and n-nonane). *Hypericum* EOs have been investigated for a wide range of biological activities in this survey: In vitro antimicrobial, antiangiogenetic, antioxidant, antiviral, antimalarial, cytotoxic, neuroprotective, tyrosinase inhibitory, and immunomodulatory activities, as well as in vivo experiments for anti-angiogenic, hepatoprotective, and wound-healing effects. However, other important aspects of the genus, such as the analysis of a standardized *Hypericum* population and the detection of new promising bioactivities, including the establishment of their mechanisms of action and potential drug interactions, should be further investigated.

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
