# Peer review of "Hypericum Essential Oils—Composition and Bioactivities: An Update (2012–2022)"

_molecules, 2022, doi:10.3390/molecules27165246_

Round 1

Reviewer 1 Report

I want to congratulate the authors, this is an interesting paper, the study is relevant, and the authors contribute to this field of research. Just a few comments I would like to illustrate:

Major comments:

1-    Follow the journal format in writing the references in the texts and match correctley with references list, e.g line 28 [4],[5] should be write as [4,5].

2-    Line 60, ‘Since then, 73 papers have been published describing relevant research 60 work, 34 of them referring to biological activities, of which ant……………etc’ need reference

3-    Table 1 and 2 add column for origin ‘i.e. add the source of collection of each species’

4-    Check the reference list and follow the journal format/style in writing.

Minor comments:

1-    In vivo and in vitro words write in the manuscript in italic form, check all texts

2-    Line 27, Convert into small letter ‘antiquity by Hippocrates [2], Dioscorides [3], and later in the medieval era by Niko laos Myrepsos

3-    Line 28 ‘WHO’ write as World Health Organization (WHO)

4-    Line 36-39 write as, ‘in folk medicine, as astringent, febrifuge, diuretic, antiphlogistic agent, analgesic, and antidepressant agents [10]. In the 18th and 19th centuries European and American physicians were used Hypericum in treatment of various health problems such as, headaches, bed wetting, burns, puncture wounds, vertigo, hyperhidrosis, melancholy, and paranoia.

5-    Line 80, ‘Results & Discussion’ write as Results and Discussion

6-    Line 162, ‘the major component on the one hand α-pinene (58% [51]; 71.2% [52]) write as, the major component on the one hand α-pinene (58%; 71.2%) [51, 52]

7-    Line 168, keep the reference number outside the brackets ‘as germacrene D (30.2% [42])…..’ to ‘as germacrene D (30.2%) [42], do it in all texts

8-    Lines 169, 171, 172 and 384, remove the italic from reference number in the text

9-    Line 203, italic ‘Hypericum’

10- Line 254, italic ‘Pirbalouti et al.

Author Response

We would like to thank the Editor and the reviewers for carefully reviewing our manuscript. We much appreciate your comments and suggestions. We have revised the manuscript accordingly. Please find the revised version entitled “Hypericum essential oils - Composition and Bioactivities: An Update (2012–2022)”, for publication as a review article in the Special Issue: Synthesis, Extraction and Biological Evaluations of Natural Products.  The comments have been addressed accordingly and highlighted (in red color) in the manuscript. The reviewer’s comments are in plain text and the response are marked in blue here below.

Reviewer 1

I want to congratulate the authors, this is an interesting paper, the study is relevant, and the authors contribute to this field of research. Just a few comments I would like to illustrate:

Answer:

Thank you for your thoughtfulness. We have revised our manuscript accordingly.

Major comments:

  • Follow the journal format in writing the references in the texts and match correctley with references list, e.g line 28 [4],[5] should be write as [4,5].

Answer:

Thank you for this comment. We corrected all reference citations as suggested.

2-    Line 60, ‘Since then, 73 papers have been published describing relevant research 60 work, 34 of them referring to biological activities, of which ant……………etc’ need reference

Answer:

We corrected the sentence accordingly eg:

‘’Ten years have passed since the last reviews have been published on Hypericum spp. EOs [17,19,20]. Since then, 66 papers have been published describing relevant research work [21-86], 34 of them referring to biological activities [20,22,24,32,34,38-42,46,49,52,54,57,58,60,62-72,74,77,79,82,83,86]..’’

3-    Table 1 and 2 add column for origin ‘i.e. add the source of collection of each species’

Answer:

Tables have been revised accordingly. We added the plant origin details in the phytochemical analysis section.

4-    Check the reference list and follow the journal format/style in writing.

Answer:

Thank you for this comment. We corrected the reference list.

Major comments:

  • In vivo and in vitro words write in the manuscript in italic form, check all texts

Answer:

We corrected the terms

2-    Line 27, Convert into small letter ‘antiquity by Hippocrates [2], Dioscorides [3], and later in the medieval era by Niko laos Myrepsos

Answer:

Thank you, we corrected it.

3-    Line 28 ‘WHO’ write as World Health Organization (WHO)

Answer:

Done

4-    Line 36-39 write as, ‘in folk medicine, as astringent, febrifuge, diuretic, antiphlogistic agent, analgesic, and antidepressant agents [10]. In the 18th and 19th centuries European and American physicians were used Hypericum in treatment of various health problems such as, headaches, bed wetting, burns, puncture wounds, vertigo, hyperhidrosis, melancholy, and paranoia.

Answer:

Thank you for this comment. We corrected the reference list.

5-    Line 80, ‘Results & Discussion’ write as Results and Discussion

Answer:

Done

6-    Line 162, ‘the major component on the one hand α-pinene (58% [51]; 71.2% [52]) write as, the major component on the one hand α-pinene (58%; 71.2%) [51, 52]

Answer:

We corrected it.

7-    Line 168, keep the reference number outside the brackets ‘as germacrene D (30.2% [42])…..’ to ‘as germacrene D (30.2%) [42], do it in all texts

Answer:

We corrected it.

8-    Lines 169, 171, 172 and 384, remove the italic from reference number in the text

Answer:

We corrected it.

9-    Line 203, italic ‘Hypericum’

Answer:

We corrected it.

10- Line 254, italic ‘Pirbalouti et al.

Answer:

We corrected it.

Reviewer 2 Report

The presented review paper adequately summarizes the data related to the composition and bioactivity of Hypericum EOs. It is comprehensive and readable, and can be of great value to researchers.

Only a few small corrections are needed in the text, which include writing the reference numbers in non-italic (lines 169, 171, 172, 267, 369, 384, 437 452, 474 which are highlighted in text), missing “%” in line 169. I would ask the authors to check through the text the uniformity of writing the term "antimalarial" (line 324), as well as in vivo and in vitro (abstract-line 17, keywords 20, lines 224, etc. italic or non-italic). Further I need to suggest authors to modify the sentence in line 49. My suggestion is: "Essential oils (EOs) represent an interesting mixture of volatile compounds, being...."

This review paper is highly recommended and these are only mirror corrections.

Author Response

Reviewer 2

The presented review paper adequately summarizes the data related to the composition and bioactivity of Hypericum EOs. It is comprehensive and readable, and can be of great value to researchers.

Only a few small corrections are needed in the text, which include writing the reference numbers in non-italic (lines 169, 171, 172, 267, 369, 384, 437 452, 474 which are highlighted in text), missing “%” in line 169. I would ask the authors to check through the text the uniformity of writing the term "antimalarial" (line 324), as well as in vivo and in vitro (abstract-line 17, keywords 20, lines 224, etc. italic or non-italic). Further I need to suggest authors to modify the sentence in line 49. My suggestion is: "Essential oils (EOs) represent an interesting mixture of volatile compounds, being...."

This review paper is highly recommended and these are only mirror corrections.

Answer:

We would like to thank the reviewer for his/her thoughtful comments and efforts towards improving our manuscript.  We have revised our manuscript accordingly.

Reviewer 3 Report

The review entitled “Hypericum essential oils - Composition and Bioactivities: An Update (2012–2022)” mainly covers the chemical composition, as well as the in vitro/in vivo pharmacological activities of the essential oils (EOs) of the genus Hypericum.

Generally, it is suitable for publication in Molecules. It will be helpful for the audience interested in this field of research.

Some minor revisions are required for the manuscript to be accepted:

1.      We know that the locations where plants are grown will dramatically affect their chemical components. It would be nice to include some site information in Table 1.

2.      The subdivision-numbered sections in section ‘3.2 Bioactivities from Hypericum essential oils’ is confusing (e.g., Lines 203 and 224, two ‘3.2 xxxx’ were presented; Line 468, 3.4 was missing). Please double-check this section and make revisions.

3.      It is suggested to add a brief discussion on the technology (e.g., GC-MS) used to define the chemical constituents of Hypericum EOs.

4.      There are typical “comma splice” mistakes in the sentences “Since then……reviews” (Lines 60-64) and “Hydro-distillation……headspace.” (Lines 93-98). Please refine them.

5.      Please double check the format of reference section. Some entries show wrong page numbers or poor consistency with others (e.g., Lines 536, 549, 567, 576, 596, 603, 615, 638, 641, 643, 662, 682, 706, 715, 719, & 729).

Author Response

Reviewer 3

The review entitled “Hypericum essential oils - Composition and Bioactivities: An Update (2012–2022)” mainly covers the chemical composition, as well as the in vitro/in vivo pharmacological activities of the essential oils (EOs) of the genus Hypericum.

Generally, it is suitable for publication in Molecules. It will be helpful for the audience interested in this field of research.

Answer:

We much appreciate your comments and suggestions. We have revised our manuscript accordingly.

Some minor revisions are required for the manuscript to be accepted:

  1. We know that the locations where plants are grown will dramatically affect their chemical components. It would be nice to include some site information in Table 1.

Answer:

Table 1 has been revised accordingly. We added the plant origin details.

  1. The subdivision-numbered sections in section ‘2 Bioactivities from Hypericum essential oils’ is confusing (e.g., Lines 203 and 224, two ‘3.2 xxxx’ were presented; Line 468, 3.4 was missing). Please double-check this section and make revisions.

Answer:

We corrected it.

  1. It is suggested to add a brief discussion on the technology (e.g., GC-MS) used to define the chemical constituents of Hypericum

Answer:

We revised the sentence accordingly eg:

‘’Nevertheless, Gas Chromatography (GC) is, by all means, the 'golden standard method' in the chemical analysis of EOs, with the aid of GC-MS (Mass Spectrometry) and GC- FID (Flame Ionization Detector), for both identification and quantification of the components and composition variations, regardless of the extraction protocol.’’

  1. There are typical “comma splice” mistakes in the sentences “Since then……reviews” (Lines 60-64) and “Hydro-distillation……headspace.” (Lines 93-98). Please refine them.

Answer:

We corrected it.

  1. Please double check the format of reference section. Some entries show wrong page numbers or poor consistency with others (e.g., Lines 536, 549, 567, 576, 596, 603, 615, 638, 641, 643, 662, 682, 706, 715, 719, & 729).

Answer:

We corrected it.
